# Emerging Protein Biomarkers for the Diagnosis or Prediction of Gestational Diabetes—A Scoping Review

**DOI:** 10.3390/jcm10071533

**Published:** 2021-04-06

**Authors:** Delia Bogdanet, Catriona Reddin, Dearbhla Murphy, Helen C. Doheny, Jose A. Halperin, Fidelma Dunne, Paula M. O’Shea

**Affiliations:** 1College of Medicine Nursing and Health Sciences, National University of Ireland Galway, H91TK33 Galway, Ireland; fidelma.dunne@nuigalway.ie; 2Centre for Diabetes Endocrinology and Metabolism, Galway University Hospital, Newcastle Road, H91YR71 Galway, Ireland; reddin.catriona@gmail.com (C.R.); dearbhlaa.murphy@hse.ie (D.M.); helen.doheny@hse.ie (H.C.D.); paulaM.OShea@hse.ie (P.M.O.); 3Divisions of Haematology, Brigham & Women’s Hospital, Boston, MA 02115, USA; jhalperin@bwh.harvard.edu

**Keywords:** gestational diabetes, biomarker, protein biomarker

## Abstract

**Introduction:** Gestational diabetes (GDM), defined as hyperglycemia with onset or initial recognition during pregnancy, has a rising prevalence paralleling the rise in type 2 diabetes (T2DM) and obesity. GDM is associated with short-term and long-term consequences for both mother and child. Therefore, it is crucial we efficiently identify all cases and initiate early treatment, reducing fetal exposure to hyperglycemia and reducing GDM-related adverse pregnancy outcomes. For this reason, GDM screening is recommended as part of routine pregnancy care. The current screening method, the oral glucose tolerance test (OGTT), is a lengthy, cumbersome and inconvenient test with poor reproducibility. Newer biomarkers that do not necessitate a fasting sample are needed for the prompt diagnosis of GDM. The aim of this scoping review is to highlight and describe emerging protein biomarkers that fulfill these requirements for the diagnosis of GDM. **Materials and Methods:** This scoping review was conducted according to preferred reporting items for systematic reviews and meta-analyses (PRISMA) guidelines for scoping reviews using Cochrane Central Register of Controlled Trials (CENTRAL), the Cumulative Index to Nursing & Allied Health Literature (CINAHL), PubMed, Embase and Web of Science with a double screening and extraction process. The search included all articles published in the literature to July 2020. **Results:** Of the 3519 original database citations identified, 385 were eligible for full-text review. Of these, 332 (86.2%) were included in the scoping review providing a total of 589 biomarkers studied in relation to GDM diagnosis. Given the high number of biomarkers identified, three post hoc criteria were introduced to reduce the items set for discussion: we chose only protein biomarkers with at least five citations in the articles identified by our search and published in the years 2017–2020. When applied, these criteria identified a total of 15 biomarkers, which went forward for review and discussion. **Conclusions:** This review details protein biomarkers that have been studied to find a suitable test for GDM diagnosis with the potential to replace the OGTT used in current GDM screening protocols. Ongoing research efforts will continue to identify more accurate and practical biomarkers to take GDM screening and diagnosis into the 21st century.

## 1. Introduction

Gestational diabetes (GDM) is defined as hyperglycemia with onset or initial recognition during pregnancy [1]. GDM is a common complication of pregnancy, with a prevalence of 5.8–12.9% globally, the prevalence varying by region, and diagnostic criteria [2]. GDM is associated with substantial short and long-term adverse outcomes for both mother and child. Short-term complications include preeclampsia and pregnancy-induced hypertension, increased risk of delivery by cesarean section, macrosomia, and neonatal hypoglycemia [3,4]. Long-term complications include increased risk of type 2 diabetes mellitus (T2DM), obesity and cardiovascular complications for both mother and offspring [5,6]. Studies have established that effective treatment of GDM reduces the rate of short-term perinatal complications and improves the quality of life of the mother [7,8]. Given this evidence, it is of utmost importance that we identify those at risk and accurately diagnose GDM [9]. Current diagnostic strategies use the oral glucose tolerance test (OGTT) performed between 24 and 28 weeks of gestation, with universal screening advised in populations with a high prevalence of T2DM [10].

As with any screening program, we must continue to re-evaluate the test suitability, accuracy, and reproducibility. The OGTT was first described in 1957 [11] and has been the gold standard for the diagnosis of GDM for decades [12]. The OGTT is onerous, lengthy and requires a fasting state [13]. A recent review by our research group [14] has detailed the numerous factors contributing to its poor reproducibility [15].

In view of the cumbersome nature and poor reproducibility of the OGTT, it is necessary to look for and identify a more robust, convenient, and accurate biomarker for the diagnosis of GDM. Over recent years, substantial progress has been made in this field of biomarkers. There is an unmet clinical need to identify an easily measurable biomarker, which is superior to the traditional OGTT. In addition, a more convenient biomarker could be used to diagnose GDM in early pregnancy, reducing the period of intra-uterine hyperglycemic exposure. This scoping review aims to synthesize the literature on emerging biomarkers for GDM diagnosis.

## 2. Materials and Methods

### 2.1. Scoping Review Question

What are the emerging biomarkers reported in the literature for the diagnosis of gestational diabetes?

### 2.2. Aim

The aim of this scoping review was to systematically identify the evidence available on emerging biomarkers with the potential to diagnose GDM (beyond glucose, fructosamine and HbA1c).

### 2.3. Methods

This review was conducted based on the framework for scoping reviews recommended by Arksey and O’Malley [16] and the later improvements to this method [17,18]. By contrast to systematic reviews, this approach was found to be more appropriate for a comprehensive search reflecting the vast number of biomarkers with a potential to diagnose GDM at the same time, enabling us to provide an in-depth analysis of selected key biomarkers [19]. Scoping reviews are a method for recording evidence from a particular research area by presenting existing research results and highlighting gaps in the evidence at the same time.

Preferred reporting items for systematic reviews and meta-analyses (PRISMA) guidelines were followed using the PRISMA extension for scoping reviews checklist [20].

No review protocol for this study has been published.

### 2.4. Data Sources and Search Strategy

Using a broad-based search strategy, the following databases were searched for relevant studies from database inception through July 2020: Cochrane Central Register of Controlled Trials (CENTRAL), the Cumulative Index to Nursing and Allied Health Literature (CINAHL), PubMed, Embase and Web of Science. Search terms used included “gestational diabetes", “GDM”, “emerging/novel/new”, “biomarkers”, “tests”, and “diagnoses” combined as appropriate using the Boolean operators “AND” and “OR” (Appendix A).

Results were inputted into the reference manager, Rayyan web application [21], and duplicates were identified and removed. Two reviewers (DB and CR) screened the titles and the abstracts. The reference lists of included studies were also reviewed. Full texts of the remaining articles were independently assessed by two reviewers (DB and CR) for eligibility based on predefined criteria. Disagreements were resolved by consensus. Where a resolution was not reached by discussion, two other reviewers were consulted (FD, POS). The electronic search strategy can be found in (Appendix A)

### 2.5. Eligibility Criteria

Studies were eligible for inclusion if study participants were pregnant women, and the study reported on a biomarker for GDM diagnosis. All study designs were eligible for inclusion. We did not apply a language restriction. However, if translation to English was not possible, the study was excluded. There was no time restriction on the date of publication of the studies. Only full-text articles were included in this review. When the full text was unavailable, the corresponding authors were contacted.

### 2.6. Data extraction and Synthesis

Data were extracted independently by two authors (DB and CR) using a standardized predetermined data collection form. For each study, we extracted the title, year of publication, journal, and biomarker (which was identified on review of the methods and results section of each paper).

Extracted data were compared for inconsistencies and merged into a final database. Disagreement was resolved through discussion and, where necessary, consultation with two further reviewers (FD, POS).

The biomarkers identified were grouped alphabetically together with all the papers citing the specific biomarker for easier identification.

It was decided that if the number of potential biomarkers identified was considerable, rendering the analysis and discussion impractical, post hoc criteria would be implemented. This would help focus the discussion on the most recent, most cited protein biomarkers.

### 2.7. Post Hoc Inclusion Criteria

Once all the biomarkers were identified, we selected for analysis and discussion biomarkers that fulfilled 3 criteria:Protein biomarkers;Biomarkers that had at least 5 citations in our search results;Study publication year: 2017–2020.

The resulting biomarkers were grouped into categories and brought forward for discussion.

## 3. Results

A total of 3519 articles were identified after the database search (Figure 1). Following title screening and deletion of duplicates, 843 abstracts were selected. A total of 458 articles were further excluded after abstract screening by two researchers, thereby reducing the articles eligible for full-paper screening to 385. A total of 53 articles were excluded (articles not in English n = 5, the test assessed was not used for GDM diagnosis n = 13, no biomarker was discussed n = 4, duplicates n = 8 and conference proceeding/abstract publication only n = 23). Finally, 332 articles were selected for data extraction. Following data extraction, a total of 589 biomarkers were identified (Appendix A).

After the application of the post hoc criteria, 15 biomarkers were identified, reviewed, and discussed. These biomarkers were grouped into 3 categories: cytokines, glycoproteins, and other proteins (Table 1). The biomarkers’ testing performance at the time of GDM and as a predictive indicator of GDM are shown in Table 2 and Table 3, and Appendix A.

## 4. Cytokines

Cytokines are cell-signaling proteins, peptides or glycoproteins that are secreted by specific cells of the immune system. They regulate and modulate both the innate and adaptive immune response to inflammation and infection.

### 4.1. Adipokines

The adipokines are cytokines secreted by the adipose tissue and comprise a group of over 600 molecules that have paracrine and endocrine functions [52]. Inflammation and dysfunction of the adipose tissue lead to a pattern of adipokines secretion, which reflects a proinflammatory, dysmetabolic and diabetogenic model [52,53].

#### 4.1.1. Adiponectin

Adiponectin is a protein secreted primarily by the fat tissue but also by the brain, the skeletal muscle, and the placenta [54,55,56], comprising 244 amino acids. Adiponectin has a role in insulin sensitivity [57,58], reduces liver gluconeogenesis [59] and enhances skeletal muscle fatty acid oxidation [60]. Low adiponectin levels are associated with an increased incidence of T2DM [61,62], and furthermore, low adiponectin levels were found in women with GDM [63,64]. This raised the question of adiponectin can be used to diagnose GDM.

Hedderson et al. [65] looked at the relationship of prepregnancy adiponectin levels and the risk of subsequent development of GDM in a case–control study within a cohort of 4098 women (GDM women n = 256, 100 g 3 h OGTT, American College of Obstetricians and Gynecologists criteria [66] controls n = 497). The team found that low adiponectin levels measured as far as six years prior to pregnancy were associated with an increased risk of developing GDM independent of age, BMI, family history or ethnicity. This finding suggests that adiponectin could have the potential to identify women at high risk of developing GDM, who otherwise would not be classified as high risk. This study, however, does not capture the changes in lifestyle, diet, and exercise between the baseline adiponectin measurement and the GDM diagnosis, and it also does not provide any information on body composition, such as percentage of fat or anthropomorphic measurements. One study has found that the first-trimester of pregnancy adiponectin is significantly lower in GDM cases compared to controls and has the potential to determine the risk of developing GDM [30] with an AUC of 0.86, thus showing promise despite the small sample size of their cohorts (n = 28). Similar results come from Williams et al. [67] and Ferreira et al. [31], who found that adiponectin levels taken at 13 weeks of gestation were lower in women, who developed GDM compared to controls. Choosing a cutoff point of 9.1 µg/mL for the first-trimester adiponectin levels, Madhu et al. [32] found the test to have a sensitivity of 100% and a specificity of 95.6% in predicting GDM.

In 2018 Bozkurt et al. [22] investigated the relationship between adiponectin levels and the development of GDM. The study included 223 participants, who were assessed for their glycemic status (75 g 2 h OGTT, IADPSG criteria) and adiponectin level at the first visit (<21 weeks of gestation) and at the second visit (24–28 weeks of gestation). The team found that adiponectin levels were significantly lower in women that developed GDM, and the association between adiponectin levels and GDM was even stronger in study participants that developed early GDM (<21 weeks), with a calculated predictive value for GDM of 0.67 (95% CI 0.57 to 0.77). Adiponectin taken during the OGTT at 24–28 weeks of gestation could predict GDM with an AUC of 0.65 (95% CI 0.57–0.74). These findings were independent of the prepregnancy maternal BMI; this is consistent with previous studies [68,69] that found adiponectin levels to be similar between individuals with a normal BMI and obese individuals that are classified as being metabolically healthy (based on lipid levels, glycemic status and blood pressure readings) compared to obese individuals classified as metabolically unhealthy. Therefore, in pregnancy, low adiponectin levels may indicate a prepregnancy predisposition for metabolic complications, such as diabetes, hypertension, or dyslipidemia, rather than a reflection on the individual’s adipose tissue mass. Weerakiet et al. [23] measured adiponectin levels in 359 women at the same time as the glucose challenge test between 21st and 27th week of gestation and, while the results were consistent with previous findings in that adiponectin level are lower in women that develop GDM independent of age and BMI, in terms of screening. However, the AUC of adiponectin was less than the glucose challenge test (GCT) AUC (0.63 (95% CI 0.53–0.67) Vs. 0.73 (95% CI 0.71–080) and had a sensitivity of 91.7% and a specificity of 30.8%. These calculations, however, were based on an arbitrarily chosen cutoff value for adiponectin at 10 µg/mL.

Xu et al. [70], in their systematic review and meta-analysis, looked at the association between adiponectin and GDM and included 15 studies and 560 GDM patients. They found that adiponectin levels were significantly decreased in women who developed GDM compared to controls, independent of BMI, similar to previous studies. The study, however, had its limitations, including large variability in adiponectin cutoff points and a high degree of heterogeneity. Iliodromiti et al. [33] conducted a systematic review and meta-analysis on the accuracy of adiponectin in predicting GDM and included 11 studies and data on 794 GDM women. They found that pooled sensitivity for adiponectin as a GDM diagnostic biomarker was 64.7% (95% CI 51%, 76.4%), and the pooled specificity was 77.8% (95% CI 66.4%, 86.1% with an AUC of 0.78 (95% CI 0.74, 0.81). While the researchers conclude that adiponectin has a moderate predictive value, there are several limitations to their paper, including the study heterogeneity, the limited access to data (2 studies), the variability of adiponectin levels cutoff points for “low” or “high” levels, the diversity of ethnicities in the populations involved and the various study designs and retrospective nature of the data that may have contributed to the results.

Adiponectin is a very promising biomarker for the diagnosis of GDM and has a significant advantage over the OGTT/GCT of not mandating a fasting state for measurements [71]. While some studies determined less than ideal performance parameters for adiponectin, we need to consider that in the first-trimester, fasting glucose has been shown to have a sensitivity of 47%, a specificity of 77% and AUC of 0.62 [72], improving in the second-trimester [73]; HbA1c has a sensitivity of 32% and a specificity of 94% [74] and fructosamine has a sensitivity of 12.2% and a specificity of 94.7% [75]. More so, there are no studies assessing adiponectin level cutoff points for best prognostic/diagnostic capacity, nor are any studies on adiponectin-trimester-specific interval ranges.

Large prospective studies together with health economic input analyzing best diagnostic cutoff points, natural level variation in GDM and normal glucose tolerance (NGT) cohorts, the impact of confounders, such as ethnicity, percentage of body fat, etc. are required to accurately determine the true value of adiponectin in diagnosing GDM.

#### 4.1.2. Chemerin

Discovered more than 20 years ago [76], chemerin (163 amino acids) is an inflammatory adipokine with a role in adipogenesis, adipocyte metabolism [77] and insulin resistance [78] secreted from the adipose tissue, liver, intestine [79] and placenta [80]. Chemerin plays a role in adipocyte metabolism, inflammation, insulin resistance and metabolic processes [77,78,81], and chemerin levels have been associated with adverse pregnancy outcomes [82,83,84].

Yang et al. [85] measured chemerin levels in the first-trimester of pregnancy (8–12 weeks’ gestation) in 212 women and in 39 women (GDM n = 19, IADPSG criteria) after the 75 g 2 h OGTT. Chemerin levels were significantly lower in the GDM group compared to NGT in the first-trimester but significantly higher in the third-trimester. In both GDM and NGT groups, chemerin significantly rose between the first and third-trimester, paralleling the rise in HOMA-IR.

In 2020 Wang et al. [24] found that the AUC of chemerin (cutoff value 6.78 µg/L) in the diagnosis of GDM (24–28 weeks of gestation) was 0.82 (95% CI 0.74–0.89) with a sensitivity of 73.3% and specificity of 76%.

Pfau et al. [86] measured chemerin levels in 40 GDM women and, while the levels were higher in GDM subjects, there was no significant difference when compared to controls; there was, however, an independent association between chemerin and markers of insulin resistance. Guelfi et al. [87] measured adipokine levels, including chemerin in 123 pregnant women at 14 and 28 weeks of gestation and found no change in chemerin concentration between the two time points and no difference in chemerin levels between women who developed GDM compared to those who did not (unlike adiponectin and leptin that showed significant changes). The cohort in this study included only women with a history of GDM with a different metabolic profile compared to the general population, so the results cannot be extrapolated. Van Poppel et al. [88] found no difference in chemerin levels between GDM (n = 15, IADPSG criteria) and NGT subjects. However, chemerin levels were significantly higher in obese women compared to non-obese women. In their systematic review and meta-analysis (10 studies), Sun et al. [89] did not find any difference in chemerin levels between GDM and NGT women but did find a positive correlation between chemerin and BMI. These results contradict the findings of Zhou et al. [90], who conducted a systematic review and meta-analysis (11 studies) looking specifically at chemerin levels and GDM and found that chemerin levels are significantly raised in the GDM population compared to NGT women. While both systematic reviews and meta-analyses had significant heterogeneity, the discrepancy in results may arise either from the different studies comprising the analysis or either from the type of subanalysis and confounders included.

The main reason for the discordant results is the fact that chemerin is influenced by numerous factors, such as inflammation, insulin resistance, metabolic syndrome, obesity, diabetes, nutrition, activity level and pregnancy [80,82,91,92,93,94,95]. It seems that chemerin may play a better role as a risk-stratifying tool rather than a GDM diagnostic biomarker, identifying women at risk of GDM, but future research might prove otherwise.

#### 4.1.3. Fetuin

Fetuins are a group of adipokines mainly secreted by the liver. Fetuin-A is secreted from the liver and adipose tissue with elevated levels in obesity [96,97], metabolic syndrome [98], fatty liver disease [99], and T2DM [100,101]. Fetuin-B, secreted by hepatocytes, tongue and placenta [102], is increased in hepatic steatosis and is linked to gluconeogenesis through insulin suppression [103,104]. Based on the association with insulin resistance and glucose metabolism, it was hypothesized that fetuins could serve as markers for GDM diagnosis.

Kansu-Celik et al. [34] measured first-trimester fetuin-A as a biomarker for GDM diagnosis in 88 pregnant women (GDM n = 29, GCT/OGTT, Carpenter and Coustan criteria) and found significantly lower levels in GDM women compared with controls. Fetuin-A below 166 ng/mL could predict GDM with a sensitivity of 58.6%, specificity of 76.2% and AUC of 0.337 (95% CI 0.21–0.46).

Kalabay et al. [105] measured fetuin-A in 134 pregnant women (GDM n = 30, 75 g 2 h OGTT, 1999 WHO criteria) and 30 non-pregnant women in each-trimester of pregnancy (including at the time of the OGTT) and found significantly higher levels of fetuin-A in GDM women at all time points compared to the NGT and non-pregnant women; fetuin-A was also positively associated with markers of insulin resistance, TNF-α and leptin levels. Iydir et al. [106] also found higher fetuin-A levels (sample collected at the time of the OGTT) in GDM women (n = 26, Carpenter and Coustan criteria [107]) compared to NGT and decreased post-partum. The authors found a positive correlation between fetuin-A and HbA1c levels. Jin et al. [35] measured fetuin-A in 270 women (GDM n = 135, IADPSG criteria) in the first and second-trimester of pregnancy and found significantly higher levels of fetuin-A in GDM women compared to controls at both time points, and it was positively correlated with the changes in the markers of insulin resistance. In this study, a fetuin-A cutoff value of 305.9 pg/mL in the first-trimester would predict GDM with a sensitivity of 64.4%, specificity of 58.5% and AUC of 0.61 (95% CI 0.54 to 0.68).

Farhan et al. [108] measured fetuin-A in 20 women (GDM n = 10) at the time of the 75 g 2 h OGTT (28 weeks of gestation) and 3 months post-partum; they found no difference in fetuin-A levels between GDM and NGT study participants at any time point.

The discrepancy between these study results arises from the different study designs, different population characteristics and sample size, and the different time-point sampling making the results inconsistent and difficult to compare.

It is unclear what the exact role of fetuin-A is in the pathophysiology of GDM. Some hypotheses suggest that its main action is through insulin resistance through the inhibition of the insulin receptor, while others suggest that fetuin-A induces adipose tissue inflammation, which leads to lipid-induced insulin resistance. There is even less information on fetuin-B, as its mode of action, signaling, and even receptor have not been adequately described. However, the minimal studies available show promising results. Prospective studies with a longitudinal sampling of both fetuins while assessing correlations with markers of insulin resistance are required.

#### 4.1.4. Leptin

Leptin (167 amino acids), the first adipokine to be discovered in 1994, [109] is predominantly secreted by adipose cells [110], but also by the stomach [111], placenta [112] and the brain [113]. Leptin has a role in energy homeostasis by inhibiting hunger and mediating food intake [114,115] through its action on the hypothalamus, dopamine system and brain stem [116]. More so, in both animal and human models, leptin administration improved hyperinsulinemia, hyperglycemia, insulin resistance and hyperlipidemia [117,118,119].

While leptin levels rise in pregnancy compared to the non-pregnant state, peaking between 20 and 30 weeks of gestation most likely secondary to fat accumulation [71,120], even higher leptin levels have been associated with GDM.

Bawah et al. [36] found that first-trimester leptin levels (11–13 weeks of gestation) in 140 women (GDM n = 70, 75 g 2 h OGTT, American Diabetes Association (ADA) criteria [121]) could predict the development of GDM with a sensitivity of 95.7%, specificity of 68.6% and AUC of 0.81.

Kautzky-Willer et al. [122] measured leptin levels at 28 weeks of gestation in GDM women (n = 55, 1999 WHO criteria [1]), women with NGT (n = 25) and women with T1DM (n = 10). These samples were collected in a fasting state and 30 minutes after the glucose load during the OGTT. They found that leptin levels were higher in women with GDM compared to NGT and T1DM and similar between NGT women and T1DM women, all matched for BMI. There were no differences between leptin levels between fasting and post-glucose-load values, indicating that this test could be done in a non-fasting state. Boyadzhieva et al. [25] measured fasting leptin levels during the OGTT in 286 women (GDM n = 127, IADPSG criteria) and found significantly higher levels in the GDM group compared to the controls. They also assessed if leptin could be used as a screening test and, setting the cutoff value at 28.7 ng/mL; the test could exclude GDM with a sensitivity of 81.2%, a specificity of 64.2% and AUC of 0.827. Bozkurt et al. [22] found higher levels of leptin in women with GDM compared to controls, but the predictive value for GDM was 0.66 (95% CI 0.57 to 0.74). Leptin taken during the OGTT at 24–28 weeks of gestation could identify GDM with an AUC of 0.61 (95% CI 0.53–0.69).

Contradictory results come from the work of McLachlan et al. [123], who found higher leptin levels in the control group compared to GDM. However, this was of borderline significance *p* = 0.05. More so, the number of women in this study was small (19 women in each arm) but well-matched, and the measurements for leptin levels were taken during an intravenous glucose tolerance test (IVGTT) in the third-trimester of pregnancy. While the OGTT is preferred over the IVGTT in detecting glucose intolerance [124], we also know from previous studies [71,120] that leptin levels peak up to 30 weeks of gestation and start decreasing thereafter.

In a systematic review and meta-analysis, Xu et al. [70] found that high levels of leptin in early pregnancy may be predictive for developing GDM independent of BMI. In their systematic review (which included 9 prospective studies), Bao et al. [125] found that leptin levels taken in the first or second-trimester of pregnancy were 7.25 ng/mL higher (95% CI 3.27–11.22) in women who were subsequently diagnosed with GDM compared to women with NGT.

The data on leptin is slightly contradictory, and that may be due to the leptin correlation with adipose tissue. Despite that, most studies show great promise. While there is some evidence that stress, sleep deprivation or exercise influence leptin levels [126,127,128], similar to the OGTT [14], the test can be done in a non-fasting state, which is a clear advantage over the OGTT. Similar to adiponectin, prospective studies are required to determine-trimester-specific reference ranges for the non-diabetic pregnant population,-trimester-specific cutoff points for the GDM population, the impact of confounders (including adiposity markers) on leptin levels and association with pregnancy outcomes.

#### 4.1.5. Omentin

Omentin (313 amino acids) is an adipose-tissue-specific factor selectively expressed in visceral tissue relative to subcutaneous adipose tissue. Omentin has a role in fat distribution, energy expenditure and insulin action modulation [129,130]. In 2007, de Souza Batista et al. [131] found that omentin levels correlated negatively with BMI/obesity, leptin level, and markers of insulin resistance and correlated positively with HDL and adiponectin levels in healthy subjects.

Barker et al. [132] studied the effects of pregnancy on omentin levels, also assessing the impact of BMI and GDM on omentin levels. Blood samples were collected in the first and second-trimester from 83 pregnant women (GDM n = 39, 75 g 2 h OGTT, Australasian Diabetes in Pregnancy Society (ADIPS) [133]). The study found significantly decreased omentin levels in non-obese GDM women compared to controls with no difference in levels between obese GDM and NGT study participants. Omentin was negatively associated with fasting glucose, and maternal BMI and no association was found between omentin and adiponectin or leptin levels. While the subgroup numbers were small and the outcomes most likely underpowered, this research was one of the first to explore the role of omentin in pregnancy and GDM, raising further questions, such as what is the balance between omentin secretion and clearance at each stage of the pregnancy; does the ratio between adipose tissue omentin secretion and placental tissue omentin secretion change during pregnancy?

Abell et al. [134] measured omentin levels in the first-trimester of pregnancy in 103 women (25 of whom later developed GDM) and found lower omentin-1 levels in women with GDM compared to controls and a negative association with 1 h and 2 h glucose levels of the OGTT. They also found that omentin-1 levels less than 38.36 ng/mL were associated with a 4-fold increased risk of GDM and that for 1 ng/mL increase in omentin levels, the risk of GDM was OR 0.97 (95% CI 0.94–0.99). Limitations of the study include that all the participants in this study were at high risk for GDM with all pregnant women being overweight or obese, and the women were initially screened with a GCT followed by the OGTT if deemed necessary with arguably milder GDM cases being missed (thus this is only attributable to the very highest risk group and not suitable for population screening). Regardless, this study also highlights that-trimester 1 omentin may have the potential to predict GDM.

Contradictory data comes from Franz et al. [135], who measured omentin levels in 192 pregnant women (GDM n = 96, German and Austrian Society for Diabetes criteria based on the hyperglycemia and adverse pregnancy outcomes (HAPO) study [136]) at the time of the OGTT, at 32 weeks and from the umbilical cord at the time of the delivery. While omentin levels were lower in the GDM group compared to the NGT group at all timepoints, this was only statistically significant at the delivery timepoint. Omentin levels were also lower in women with a higher BMI and a lower HDL cholesterol.

In a systematic review and meta-analysis, which included 20 studies (GDM n = 1493), Sun et al. [89] found that omentin levels were significantly lower in women with GDM than in healthy controls. The authors also suggested that age and BMI may be important parameters influencing omentin levels in GDM patients. While there was significant heterogeneity in this review and a limited number of studies identified, the authors conclude that omentin has the potential to be a novel biomarker for early GDM diagnosis.

Omentin shows some promise as a GDM diagnostic biomarker, however further studies are required to clarify the actual role in GDM pathophysiology—if it is linked to visceral adiposity, vascular/endothelial dysfunction in either visceral adipose tissue or placenta or insulin mediation. Prospective studies are required to detect specific reference ranges and cut-offs and assess the impact of adiposity and inflammation on omentin levels and consecutively on omentin capacity to diagnose GDM. More so, omentin levels are influenced by fasting state [137,138], which makes it a less attractive biomarker compared to other biomarkers discussed.

#### 4.1.6. Interleukin 6 (IL-6)

IL-6 is an inflammatory cytokine [139] secreted by monocytes/macrophages, but also endothelial cells, myocytes, adipocytes, pancreatic cells and placenta [139,140] with primary roles in immune response regulation, inflammation and hematopoiesis [141], but also roles in obesity, insulin resistance and T2DM [142,143,144]. Some of the mechanisms proposed for its role in metabolism are the percentage of body fat [145], the degree of visceral fat [146], IL-6 direct effect on hepatocytes [142,147], the immune response induced dyslipidemia, IL6 lipolytic effect [148] or even a central effect of IL-6 on food intake [149,150]. A systematic review and meta-analysis by Liu et al. (Liu 2016) explored the association between IL-6 and T2DM. It comprised 16 studies involving 24,929 subjects and found that IL-6 was a strong predictor of developing T2DM. In pregnancy, the role of IL-6 in GDM prediction has given conflicting results.

Sudharshana Murthy et al. [151] explored the role of IL-6 in GDM. IL-6 levels were taken at the time of the OGTT in 60 pregnant women (GDM n = 30, OGTT) and found significantly raised IL-6 levels at the time of diagnosis. Siddiqui et al. [152], using a very similar study design, measured IL-6 levels in 103 pregnant women (GDM n = 53, OGTT, ADA criteria) at the time of the OGTT and found significantly increased IL-6 levels in the GDM cohort compared to the NGT and a strong association between IL-6 levels and prepregnancy BMI and fasting and post-prandial glucose levels. The participants in both studies were Asian with a median normal/normal-high BMI. A prospective study by Braga et al. [153] involving 176 South American pregnant women (GDM n = 78, 100 g OGTT, Carpenter and Coustan criteria) found no difference in IL-6 levels (taken at the time of the OGTT) between GDM and NGT women. Similar results were found by Simjak et al. [154] in 24 European pregnant women (GDM n = 12, OGTT, IADPSG criteria) with normal BMI, who examined IL-6 levels in the second and third-trimester and post-partum and found no difference between GDM and NGT women.

Driven by the discordance in results, a recent systematic review by Amirian et al. [155] has explored the relationship between Il-6 and GDM in studies published between 2009 and 2020 and included 24 articles. The study highlighted the diversity of ethnicities involved, the different measurement methods, but also the numerous criteria used to diagnose GDM (14 different diagnostic criteria) in the studies selected, making significant research synthesis difficult. The common denominator for all studies, however, was the small sample size with the largest cohort in a study by Abdel Gader et al. [156], who found no difference in IL-6 levels between GDM and NGT women. Out of 24 studies, 16 found a positive association between IL-6 levels and GDM, the authors concluding that IL-6 can be used as a GDM biomarker. However, such a statement requires more scientific evidence. The heterogeneity of the studies to date involved in assessing the relationship between IL-6 and GDM is too high to be able to make any meaningful comparison.

Conceptually, IL-6 could be linked to GDM pathogenesis either through a higher degree of inflammation in GDM pregnancies [157], driven by increased subcutaneous or visceral adipose tissue [158] or increased IL-6 secretion by the placenta in GDM pregnancies [159]. While IL-6 might prove to be a good GDM biomarker in the future, there are too many unanswered questions at present for such a claim. Larger studies with increased homogeneity in GDM diagnostic methods and criteria are required with serial IL-6 measurements in each-trimester of pregnancy for identification of-trimester-specific ranges, measurements of subcutaneous and visceral adipose tissue, which might be the driver for its increase, and associations with adverse pregnancy outcomes.

#### 4.1.7. Tumor Necrosis Factor (TNF)

TNF is an inflammatory cytokine family primarily secreted by monocytes/macrophages [160] with two main components TNF-α (also secreted from the placenta [161]) and TNF-β. The initial role for TNF was thought to be the death of tumor cells [162], but it was soon discovered that TNF plays an important role in inflammatory diseases [163], neurodegenerative disease [164], and depression [165]. Given its pro-inflammatory effects, TNF has been identified as a marker of metabolic syndrome [166,167], obesity [168] and insulin resistance [169,170]. Evidence suggests that TNF stimulates the secretion of IL-6 [171,172], inhibits the secretion of adiponectin [173], induces apoptosis in adipose cells [174,175] and inhibits the insulin receptor, thus promoting insulin resistance [170,176]. A recent study by Alzamil et al. [177] examined the correlation between TNF-α and insulin resistance, T2DM and obesity in 128 Asian subjects (T2DM n = 65). These authors found significantly higher TNF-α in T2DM subjects compared to controls, in obese subjects (T2DM or non-T2DM) compared to non-obese subjects, and TNF-α levels were positively correlated with HbA1c levels and HOMA-IR highlighting the role TNF-α plays in the pathogenesis of insulin resistance and T2DM and the link with both obesity and glucose intolerance.

Guillemette et al. [178] studied TNF-α levels in both the first-trimester of pregnancy and at the time of GDM diagnosis and its relationship to GDM in 756 pregnant women (GDM n = 61, GCT/OGTT, IADPSG criteria). They found a positive association between TNF-α levels and BMI, adiponectin, and insulin levels in the first-trimester and HOMA-IR, BMI, triglycerides, and fasting insulin levels in the third-trimester. The authors also showed that TNF-α levels are strongly positively linked to insulin resistance and that it behaves differently during the OGTT in insulin-sensitive and insulin-resistant women.

Kirwan et al. [179] described longitudinal changes in TNF-α levels and the association with maternal insulin resistance in 15 women (GDM n = 5, euglycemic-hyperinsulinemic clamp, Carpenter and Coustan criteria). They found that TNF-α in normal pregnancy had lower levels in early pregnancy, increasing in late pregnancy paralleling insulin sensitivity changes, with higher levels in GDM women compared to lean NGT women. They also found that TNF-α was positively correlated with insulin sensitivity independent of BMI or glycemic status. Proposed mechanisms for this were either increased TNF-α secretion by the placenta in GDM women and direct inhibition of the insulin receptor. This hypothesis is also supported by Desoye et al. [159]. Syngelaky et al. [37] studied the link between first-trimester TNF-α and the development of GDM in 1000 women (GDM n = 200, random glucose/OGTT, WHO criteria) and found higher TNF-α levels in women with GDM compared to controls. The authors calculated that TNF-α could predict GDM development with an AUC of 0.82, but adding TNF-α levels to a multi-variable prediction model did not improve any of the estimated variables. While this was a large study, the GDM diagnostic method may have omitted milder cases of GDM that could have been included in the NGT group. A recent study by Wang et al. [24] explored TNF-α levels at GDM diagnosis in 110 Chinese pregnant women (GDM n = 60, OGTT, ADA 2017 criteria) and found significantly higher TNF-α levels in GDM women compared to controls.

No correlation between TNF-α levels (samples were taken in both first-trimester and at GDM diagnosis) and GDM development was found by Georgiou et al. [30] in 250 women (GDM n = 14, OGTT, ADIPS criteria).

A systematic review and meta-analysis by Xu et al. [70] on the association between GDM and TNF-α levels comprised 10 studies, and despite the increased heterogeneity of the studies and missing confounders from the analysis, the authors found overall significant high levels of TNF-α in GDM pregnancies compared to controls independent of BMI.

The discrepancy in study results most likely lies in the different sample sizes, ethnicities, diagnostic methods and criteria and concentration limits employed. In addition, fasting, exercise and stress influence TNF-α levels [180,181,182,183,184], and this needs to be taken into account when considering new diagnostic tests. While there is no doubt that TNF-α plays a role in the pathogenesis of GDM and insulin resistance, the actual predictability value of this biomarker is yet to be established.

### 4.2. Glycoproteins

#### 4.2.1. Afamin

Afamin is a glycoprotein present in plasma, cerebrospinal fluid, ovarian and seminal fluid [185,186], primarily expressed in the liver, but also expressed in the brain and kidneys and its main role to bind and transport vitamin E [185,187] to peripheral tissues and organs. Studies examining afamin levels in polycystic ovary syndrome (PCOS) cohorts, despite a relatively small sample size, have found an association between afamin and insulin resistance and metabolic syndrome [188,189]. In a large multicenter study (n = 20,136), Kollerits et al. [190] found that afamin was a strong predictor for the development of T2DM and strongly correlated with insulin levels, HOMA-IR and insulin resistance, suggesting that afamin has the potential to be a biomarker for early prediction for future development of T2DM.

In pregnancy, afamin levels raise progressively with each-trimester of pregnancy, decreasing back to baseline post-partum with even higher levels in pregnancies complicated by preeclampsia or hypertension [191]. Based on the previous findings, which linked afamin with the development of insulin resistance and diabetes, it has been hypothesized that afamin may serve as a predictor for GDM. In two studies, Tramontana et al. [38,192] explored the relationship between first-trimester afamin and pregnancy complications in 4948 pregnant women and found significantly higher levels of afamin in women who subsequently developed GDM (n = 207, IADPSG criteria) compared to NGT women. Afamin (cutoff value > 65 mg/L) was shown to be an independent predictor for developing GDM with a risk ratio of 2.07 (95% CI 1.33–3.22) and an AUC of 0.66. Koninger et al. [39] looked at prepregnancy afamin levels in predicting GDM in a PCOS population (n = 63, GDM n = 29) and found higher afamin levels and HOMA-IR in women who developed GDM compared to controls with a strong positive correlation between afamin and HOMA-IR. The team showed that an afamin level of 88.6 mg/L identified GDM patients with a sensitivity of 79.3%, specificity of 79.4% and an AUC of 0.78 (95% CI 0.65–0.90). Ravnsborg et al. [40] studied potential GDM biomarkers in 270 first-trimester samples with shotgun proteomics (GDM n = 135), diagnosed according to the Danish guidelines [193] and found higher afamin levels in GDM women compared with controls and that afamin could predict GDM diagnosis with an AUC of 0.67 (95% CI 0.53–0.81).

Koninger et al. [194] studied the predictive power of afamin in diagnosing GDM in both the first-trimester (n = 110, of which 59 developed GDM) and the second-trimester of pregnancy (n = 105, of which 29 developed GDM). GDM was diagnosed according to the German Diabetes Association (DDG) and the German Association for Gynecology and Obstetrics (DGGG) [195]. They found that both first and second-trimester afamin levels were higher in GDM women compared to NGT. Because this study comprised two different cohorts for first and second-trimester samples, the samples were not taken longitudinally. Therefore, the team was not able to determine-trimester-specific cutoff values for afamin levels. Another limitation of the study is the heterogeneity of the GDM diagnosis method and criteria used as not all women were screened with 75 g 2 h OGTT, and milder cases of GDM might have been missed.

Afamin is a very novel biomarker for GDM. It is not fully clear what is the exact mechanism through which afamin is linked to insulin resistance, metabolic syndrome, and glucose intolerance. In previous studies [186,196], there has been observed no variation in afamin levels between fasting and non-fasting state, no circadian variation, no variation with the menstrual cycle or gender variation, suggesting that afamin is a stable biomarker for longitudinal measurements. There is not enough evidence to clearly state the true potential of afamin in predicting GDM, but the results to date are promising.

#### 4.2.2. CD59

CD59 is an 18–20 kDa glycoprotein, which is also known as membrane attack complex (MAC) inhibitory protein (MAC-IP) [197,198]. Its main role is to restrict MAC formation in the cell membrane, thus preventing cell lysis and cell death. While CD59 is a protein bound to the cell membrane, soluble forms are present in the blood, urine, and saliva [199,200,201].

The link between diabetes complications and increased MAC deposits has been well documented [202,203,204,205,206,207]. The first paper linking the increased MAC deposits in diabetes with CD59 inactivation was by Acosta et al. [208]. They showed that in vitro CD59 exposure to glucose reduced its protection role leading to cell lysis. Building on this work, Qin et al. [207] measured CD59 levels in the red blood cells (RBC) of subjects with and without T2DM and found that there are significantly lower levels of CD59 in diabetic RBC compared to subjects without T2DM.

In 2013, Ghosh et al. [209] hypothesized that glycated CD59 (gCD59) levels might mirror glucose control in human subjects and developed a sandwich ELISA assay to identify plasma gCD59, which they tested initially in 24 participants with and without T2DM (T2DM n = 14 HbA1c > 48 mmol/mol) and then validated it in 190 subjects (T2DM n = 100). gCD59 levels were significantly higher in the 14 individuals with T2DM from the initial testing set compared to controls and were strongly associated with HbA1c levels. gCD59 was able to identify T2DM with a sensitivity of 93%, specificity of 100% and AUC of 0.98. In the follow-up testing set, gCD59 levels were indeed higher in the T2DM group and positively associated with HbA1c levels, with the test generating an AUC of 0.88. Continuing this work, Ghosh et al. [210] explored the link between gCD59 and glycemic variables, such as HbA1c (in 400 subjects (T2DM n = 226) and glucose levels during the OGTT (n = 109). The results supported previous findings, with gCD59 levels higher in diabetic vs. participants without diabetes and independently associated with HbA1c and with the 2 h glucose level on the OGTT. More so, the team also showed an acute response of gCD59 levels to insulin therapy in 21 poorly controlled subjects, with changes in levels in 2 weeks of treatment, while HbA1c and fructosamine took 6–8 weeks to respond. This rapid turnover of values would have particular importance in pregnancy and GDM where time is limited and in utero exposure to hyperglycemia not without consequences.

Ma et al. [27] studied gCD59′s capacity to predict GDM earlier in pregnancy (sample collected and OGTT performed <20 week’s gestation) and the association with adverse pregnancy outcomes using 770 frozen samples collected as part of the vitamin D and lifestyle intervention (DALI) study (Simmons D 2017). All the participants in the DALI study had a BMI ≥ 29 kg/m^2^ and underwent 3 OGTT s (<20 weeks GDM n = 207, 24–28 weeks GDM n= 77 and 35 weeks of gestation) and were diagnosed according to the IADPSG criteria. gCD59 levels were higher in GDM women diagnosed <20 weeks of gestation independent of age, BMI or ethnicity and predicted the OGTT results <20 weeks with an adjusted AUC of 0.86 (95% CI, 0.83–0.90). Restricting the analysis to the OGTT performed between 14 and 20 weeks of gestation, the AUC was calculated at 0.90 (95% CI 0.86–0.93). Early gCD59 predicted GDM at 24–28 weeks with an AUC of 0.68 (95% CI 0.64–0.73). The team also found that higher gCD59 levels were associated with the risk of delivering an LGA baby. Some limitations of the study include the retrospective nature of the study, the inclusion of only high-risk women with a BMI ≥ of 29 kg/m^2^ and low ethnic diversity.

In 2017, Ghosh et al. [26] explored the association between gCD59 and the results of the GCT, the results of the OGTT and the prevalence of large for gestational age (LGA) babies in 1000 pregnant women at 26 weeks of gestation (500 women passed the GCT and were controls and 500 women failed the GCT and underwent a 3 h OGTT). gCD59 was 8.5 times higher in women who failed the GCT compared to those who passed it and 10 times higher in women who were diagnosed with GDM (n = 127) on the 3 h OGTT (Carpenter and Coustan criteria). gCD59 predicted GCT failure with a sensitivity of 90%, specificity of 88% and adjusted AUC of 0.92 (95% CI 0.88–0.93) and predicted the development of GDM compared to controls with a sensitivity of 85%, specificity of 92% and adjusted AUC of 0.92 (95% CI 0.77–0.91), independent of age, BMI, ethnicity of history of diabetes. More so, the team also identified significantly higher gCD59 levels in women who gave birth to an LGA baby. There are some limitations to this study, including its observational nature, the use of GCT (which may not be done in the morning), the 3 h-OGTT and the Carpenter and Coustan criteria for GDM diagnosis arguably missing milder cases of GDM.

gCD59 is a very promising biomarker that has shown much potential in the diagnosis and early diagnosis of GDM and the prediction of LGA-born infants. The rapid turnover of values and the lack of need for fasting certainly is a significant advantage for a pregnancy biomarker. However, there are still unanswered questions, such as: what are the-trimester-specific cutoff values? Are there any discrepancies in cutoff values among different ethnicities? Could early pregnancy gCD59 predict the 24–28 OGTT results in a BMI diverse population? Larger prospective studies are required to answer these questions, and one such study is currently underway [211].

#### 4.2.3. Human Chorionic Gonadotropin (hCG)

HCG is a glycoprotein hormone, mainly secreted by the placenta, whose main role is in embryo implantation and control of embryogenesis [212]. Recently, however, Ma et al. [213] have shown that hCG influences insulin sensitivity and induces adipocyte-mediated inflammation and consequently may contribute to GDM pathogenesis. The beta isoform of hCG (β-hCG) is part of the first-trimester screening for fetal aneuploidy.

Sirikunalai et al. [214] retrospectively studied the link between β-hCG levels and adverse pregnancy outcomes, including GDM in 13,620 pregnant Thai women and found that high first-trimester β-hCG levels were associated with a decreased risk of developing GDM. This finding was not sustained in the second-trimester. While this study had a large sample size, a high number of women had incomplete data, and due to the retrospective nature of the study, adequate adjustments and multivariate analysis could not be done due to the lack of absent confounders. Ong et al. [215] measured β-hCG levels between 10 and 14 weeks of gestation in 5584 pregnant women. Women were diagnosed with GDM with a 2 h OGTT and diagnosed according to the 1980 WHO criteria [216]. The team found significantly lower β-hCG levels in women that developed GDM (n = 49) compared to NGT, suggesting that first-trimester β-hCG could predict second-trimester GDM diagnosis. A limitation of this study and an explanation for the small number of GDM cases detected is the GDM diagnosis criteria used, which would only identify severe cases of GDM, with milder cases not being included in the study. The use of IADPSG criteria in this cohort would have led to a more representative sample of the general population and ease the generalizability of results. Xiong et al. [217] retrospectively analyzed β-hCG levels in 1596 cases, 11 days after single blastocyst transfers (assisted reproduction) with 370 live births and found significantly higher rates of GDM (GDM total n = 61) in women with low levels of β-hCG compared to women with high levels of β-hCG. Beyond the retrospective nature of the study, no information is provided on the GDM diagnosis and criteria used; the number of GDM women in the low-level β-hCG subgroup is quite small (n = 5) and insufficient for a robust comparison. Controversially, Yue et al. [218] measured β-hCG levels between 14 and 20 weeks of gestation in 8333 pregnant Asian women, 1336, of which developed GDM (ADA criteria) and found high β-hCG levels are an independent risk factor for the development of GDM. A possible explanation for this discrepancy may be the more advanced week of gestation when the sample was collected with reactive β-hCG levels secreted by a hypoxic placenta as a response to hyperglycemia; also, the overall BMI of the cohort was very low compared to previous studies.

In a retrospective study, Tul et al. [219] measured first-trimester β-hCG levels in 1136 Caucasian women (GDM n = 27) and found lower yet not statistically significant levels in women who developed GDM. In this cohort, GDM was diagnosed with the 3 h OGTT and given the number diagnosed, and it would equate to a GDM prevalence of 2.37%, which is extremely low compared with the overall European GDM prevalence. It is unclear from the paper whether this low prevalence is due to missing data. However, these results are supported by Savvidou et al. [220], who retrospectively assessed β-hCG levels at 11–13 weeks of gestation in 42,102 pregnant women. GDM (n = 779) was diagnosed with a 2-step approach, the women undergoing an OGTT only if the random plasma glucose at 24–28 weeks of gestation was higher than 6.7 mmol/L. The team found no difference in β-hCG levels between women who developed GDM and NGT. No correlation in first-trimester β-hCG levels and GDM development was also found in Beneventi et al. [221] (GDM n = 228, GCT/ 100 g 3 h OGTT, Carpenter and Coustan criteria) or Sweeting et al. [222] (GDM n=248, OGTT, Australian Diabetes in Pregnancy Society criteria [133]).

There is quite a high degree of heterogeneity in design, populations and GDM diagnosis methods leading to inconsistent results. None of the studies looked at the longitudinal trend of β-hCG levels in the first and second-trimester and GDM diagnosis, which would have clarified the cause of variable levels—low levels in the first-trimester secondary to compromised placentation or reduced placental mass; high levels in the second-trimester of pregnancy secondary to hyperglycemia-induced placental hypoperfusion. It seems, however, that studies, which involved a higher-risk population for the development of GDM (assisted reproduction, GDM diagnosis criteria that identifies more severe cases of GDM, etc.) were more likely to find an association between β-hCG levels and GDM. Perhaps single β-hCG levels could be used to identify a possible at-risk GDM population that should be adequately followed up and screened. However, current evidence does not support this, and future more consistent studies are required.

#### 4.2.4. Sex-Hormone Binding Protein (SHBG)

SHBG is a glycoprotein produced by the liver, brain, uterus, testes and placenta [223], and its main role is to bind and transport biologically active androgens and estrogens [224]. SHBG is linked to adipose tissue with lower levels in obese subjects [225], which increases when weight loss is achieved [226,227]. SHBG also has been linked to insulin resistance [228,229], metabolic syndrome [230,231,232] and the development of non-alcoholic fatty liver disease (NAFLD) independent of BMI and T2DM [233,234,235]. Potential mechanisms suggested for this are either a direct effect of insulin on SHBG production [236,237] or fat accumulation in the liver and/or increased hepatic triglycerides levels leading to decreased SHBG gene expression [238,239,240]. The role of SHBG in GDM diagnosis has been explored in numerous studies at different time points during pregnancy, including prepregnancy, with overall promising results.

Veltman-Verhulst et al. [43], in a prospective study, measured SHBG in 50 women with PCOS prior to pregnancy (median 35 weeks) following fertility treatment. GDM diagnosis was made based on a 3 h OGTT at 24–28 weeks of gestation (GDM n=21). SHBG levels were significantly lower in the GDM group compared to NGT and, with a cutoff level of 58.5 nmol/L, SHBG could predict GDM with a sensitivity of 81.%, specificity of 82.8% and AUC of 0.86 (95% CI 0.75–0.97). Hedderson et al. [241] studied the link between prepregnancy (median 6.2 years) levels of SHBG and the subsequent development of GDM in a case–control study (GDM n = 267, 3 h OGTT, Carpenter and Coustan criteria) and found a significantly lower level of SHBG in women, who developed GDM independent of GDM risk-factors. This study showed that SHBG levels measured years (min. 6 years) prior to pregnancy could predict the development of GDM even in very low-risk women, and this is of high clinical importance. Study limitations include the lack of longitudinal anthropomorphic data, additional SHBG measurements during pregnancy and the lack of markers of visceral adiposity. Badon et al. [44] measured SHBG in a case–control study within a cohort of 4098 pregnant women (GDM n = 267) at a median 7 years prior to pregnancy and similar to previous studies found significantly lower SHBG levels in women who subsequently developed GDM compared to controls with a predictive value of 0.71.

In a longitudinal study, Li et al. [242] measured SHBG levels in 321 women (GDM n = 107) in all 3-trimesters of pregnancy. SHBG levels increased progressively with the-trimester of pregnancy in both GDM and NGT groups, with significantly lower levels in GDM compared to controls in the first-trimester. This significance disappeared in late pregnancy, suggesting that perhaps the best time to measure SHBG is early in pregnancy as lifestyle changes or treatment for GDM in late pregnancy may influence SHBG levels. They also found SHBG levels to be negatively associated with markers of insulin resistance. This study confirmed the results of a previous study by Smirnakis et al. [243], who measured SHBG levels in 145 women (GDM n = 37, GCT, ACOG criteria [244]) at 11 and 17 weeks of gestation and found lower levels in women, who subsequently developed GDM compared to controls, with the stronger association at 11 weeks of gestation. Caglar et al. [41] found that an SHBG cutoff level of 97.47 nmol/L (at 13–16 weeks of gestation) could predict GDM with a sensitivity of 46.7%, specificity 84.1% and AUC 0.67 (95% CI 0.55–0.79), while Maged et al. [42] using a first-trimester SHBG cutoff value of 211.5 nmol/L calculated a sensitivity of 85.2%, specificity 37% and AUC of 0.69.

Tawfeek et al. [28], in a case control study, measured SHBG levels at the time of the OGTT at 24–28 weeks of gestation (GDM n = 45, 75 g 2 h OGTT, IADPSG criteria) and found significant lower SHBG levels in the GDM group compared to NGT. At a cutoff value of 50 nmol/L, SHBG could identify GDM with a sensitivity of 96%, specificity of 95% and AUC of 0.91 (95% CI 0.82–1.) A similar study in the same population by Siddiqui et al. [245] (GDM n = 53, OGTT, ADA criteria) also found lower levels of SHBG in GDM women compared to controls, but only in nulliparous women with a positive correlation with gestational age. Limitations of both studies include the small sample size with a relatively high BMI and ethnically confined to Asian participants, which tend to have a higher prevalence of metabolic syndrome [246].

McElduff et al. [247], however, found no difference in SHBG levels between GDM and NGT groups in their cross-sectional study, which included 220 pregnant women (GDM n = 642, GCT/OGTT, Carpenter and Coustan criteria). Despite a robust study methodology similar to previous studies, the main reason for these discordant results lies in the difference in population characteristics and diagnostic method.

SHBG is a straightforward, low-cost test that does not require fasting [248] and has no diurnal variation [249]. This test has shown some promise as a predictor of GDM when used prior to or in the first-trimester of pregnancy, and this might be because the difference in insulin resistance markers reduces as the pregnancy progresses. Catalano et al. [250] found higher levels of insulin resistance in the first-trimester of pregnancy in women with NGT in the first-trimester, who eventually developed GDM compared to women with NGT all throughout. In most studies, the link between SHBG and GDM was independent of subcutaneous adiposity, suggesting, perhaps, that liver adiposity would be a better marker for SHBG production [251]. However, further studies are required to measure SHBG levels, markers of insulin resistance, glycemic control and hepatic steatosis for a better understanding of the role of SHBG in GDM. Beyond that, however, we cannot ignore the positive results in studies assessing prepregnancy and first-trimester predictability of GDM development, which would allow for early interventions on the modifiable factors involved in GDM development. Standardization of prepregnancy and-trimester-specific cut-offs, GDM diagnosis cutoff value, exploration of the variability of levels in different populations and standardizations of assays would make SHBG a very promising marker for the early diagnosis of GDM.

### 4.3. Other Proteins

#### 4.3.1. C-Reactive Protein (CRP)

CRP is an acute-phase protein secreted and released by numerous cells in the context of inflammation [252,253]. Obesity, which is a proinflammatory state, is a known risk factor of GDM. CRP is a nonspecific marker, which may be elevated in settings, such as infection or obesity, in the absence of GDM [254,255,256]. High levels of CRP have been described in the association with insulin resistance and metabolic syndrome [257,258,259,260]. Numerous studies have found an association between obesity and high CRP levels independent of insulin resistance [261,262,263]. Therefore, it is biologically plausible that inflammatory markers, such as C-reactive protein (CRP), could be a promising biomarker of GDM.

Alamolhoda et al. [264] prospectively studied the relationship between first-trimester CRP levels and the risk of developing GDM in 120 pregnant women (GDM n = 11, OGTT) and found a significant difference between GDM women and controls independent of BMI. The sample size, however, was small, and the cutoff value for fasting glucose at diagnosis was 7 mmol/L (126 mg/dL), which selected only the more severe cases, fasting glucose of 7 mmol/L being the cutoff diagnostic value for T2DM. In a case–control study involving 372 women (GDM n = 124, OGTT, WHO criteria), Savvidou et al. [265] also found higher first-trimester CRP levels in women who subsequently developed GDM compared to controls. While the sample size was larger, similar to the previous study, the fasting glucose cutoff was 7 mmol/L (126 mg/dL). Kansu-Celik et al. [34] investigated first-trimester high-sensitivity CRP (hsCRP) as a biomarker for GDM diagnosis in 88 pregnant women (GDM n = 29, GCT/OGTT, Carpenter and Coustan criteria) and found that hsCRP was significantly higher in women, who subsequently developed GDM independent of BMI with a sensitivity of 86.2%, specificity of 50.8% and AUC of 0.70 (95% CI 0.59–0.81) at a cutoff value for hsCRP of 4.65 ng/mL. Wolf et al. [266] measured first-trimester CRP levels in 131 women (GDM n = 43, GCT/OGTT, ADA criteria) and found that women diagnosed with GDM had higher CRP levels in the first-trimester of pregnancy with a strong positive correlation between CRP levels and 1 h post glucose load levels and systolic blood pressure. The team also found that the addition of BMI in a multivariate model attenuated the correlation between CRP and GDM diagnosis suggesting the influence of adipose tissue on CRP levels. Some of the limitations of this study include the small sample size, the single time point measurement of CRP and the lack of additional adiposity markers beyond BMI (such as waist circumference, visceral fat).

Alyas et al. [267] measured high-sensitivity CRP (hsCRP) levels at 14–18 weeks of gestation and 24–28 weeks of gestation in 158 women (GDM n = 58, OGTT, IADPSG criteria) and found significantly higher CRP levels at both time points in women diagnosed with GDM compared to controls. No analysis on BMI category was done in this study.

Conflicting results are found in a cross-sectional study by Korkmazer et al. [268], who measured hsCRP at GDM screening time point in 116 women, who underwent a GCT followed by an OGTT and were classified in GDM (n = 39, failed GCT and OGTT, Carpenter and Coustan criteria) and glucose intolerant (n = 37, abnormal GCT, normal OGTT) and controls. The team found no differences in hsCRP levels between the three groups. The sample size was small, which may explain these findings, and no analysis was done between the GDM and glucose intolerance groups together and the controls. Corcoran et al. [269] evaluated hsCRP in 225 pregnant women with one or more risk factors for GDM in the first-trimester (46, of which developed GDM, OGTT, IADPSG criteria) and found no difference in hsCRP levels between the GDM and the control group findings also supported by Adam et al. [270]. Retnakaran et al. [255] measured CRP levels at the time of the OGTT in 180 women (GDM n =39, impaired glucose tolerance n = 48, GCT/OGTT, National Diabetes Data Group (NDDG) criteria [271]) and found no association between CRP levels and glycemic pregnancy status, but did describe a strong association between CRP levels and prepregnancy BMI and fasting glucose.

A systematic review by Amirian et al. [272] investigated the association between CRP and GDM diagnosis and included 31 articles. Even though no meta-analysis was done in this study due to the lack of clinical data, the authors found a positive association between high CRP levels and GDM development in 20 studies (CRP n = 8 articles, hsCRP n = 12 articles), while 11 studies (CRP n = 6 articles, hsCRP n = 5 articles) did not identify any correlation. The main reasons for these discrepancies are the variations in diagnostic methods and criteria, the difference in sample size and population characteristics, different methods to quantify CRP levels or the lack of adjustment for BMI and other confounders.

While CRP/hsCRP shows some potential, the literature shows inconsistent and contradictory data, with most studies having small sample size cohorts. This arises from the wide arrays of methodology and study population features. A big disadvantage of using CRP as a diagnostic biomarker is its nonspecificity as a high result will possibly lead to a wide range of investigations, some unnecessary, increasing costs and the pregnant woman’s stress levels. Further research is required to clarify the correlation between adiposity (subcutaneous or visceral) and CRP levels and the reflection of this association in glycemic status. CRP may play a more meaningful role as a risk assessment tool for GDM screening rather than GDM diagnosis.

#### 4.3.2. Nesfatin-1

Nesfatin-1, initially described in 2006, is a neuropeptide produced primarily by the hypothalamus and brain stem, and its main role is in food and water intake regulation, control of appetite with anorexigenic properties [273]. Research on animal models [274,275] found that intravenous nesfatin-1 regulated fatty acid metabolism, reduced insulin levels, improved insulin sensitivity and reduced blood glucose levels in mice. Li et al. [276] measured nesfatin-1 levels in healthy adults and adults with T1DM and T2DM and found significantly lower nesfatin-1 levels in individuals with T2DM compared to controls (not valid for T1DM) independent of BMI and no significant change in levels during the OGTT. A systematic review and meta-analysis by Zhai et al. [277] studied the association between T2DM and nesfatin-1 and comprised 7 studies and 627 participants (T2DM n = 328), with 6 out of 7 studies being carried out in China. The authors found significantly higher nesfatin-1 levels in newly diagnosed T2DM compared to controls; however, overall, when all participants were included, there was no significant association between T2DM and nesfatin-1. All the studies included had small size numbers of participants, and there was no subanalysis on the duration of diabetes, BMI subcategory, insulin resistance markers or glycemic control (HbA1c) on or off treatment.

Given the anorexigenic, the antihyperglycemic effect of nesfatin-1 and moderate evidence of the association with T2DM, studies have also assessed a possible implication of nesfatin-1 in GDM. A prospective study by Kucukler et al. [278] measured nesfatin-1 levels at 24–28 weeks of gestation in 79 pregnant women (GDM n = 38, GCT/OGTT, ADA criteria) and found significantly lower nesfatin-1 levels in the GDM group compared to NGT at diagnosis. Nesfatin-1 was negatively associated with BMI, fasting glucose, and HOMA-IR. Measurements at 24–28 weeks of gestation were also taken by Ademoglu et al. [279] in 70 pregnant women (GDM n = 30, GCT/OGTT, Carpenter and Coustan criteria) and similar to previous studies found lower nesfatin-1 levels in women with GDM compared to NGT women independent of age, BMI, fasting glucose and HOMA-IR at diagnosis. No correlation was found between nesfatin-1 and fasting glucose, BMI, or markers of insulin resistance. In a larger study, Mierzynski et al. [280] measured nesfatin-1 levels at 24–28 weeks of gestation in 237 women (GDM n = 153, OGTT, WHO criteria) and found lower nesfatin-1 levels in GDM subjects compared to controls. There was a positive association between nesfatin-1 levels and BMI, glucose levels and gestational age.

Nesfatin-1 is a recent biomarker with only a few small size studies exploring its role in GDM pathogenesis. It is a possibility that nesfatin-1 could act as an antidiabetic agent by enhancing insulin action/ secretion, reducing glucose levels, and reducing food intake, and a decrease in nesfatin-1 levels in pregnant women may lead to insulin resistance and GDM. Another theory is that insulin resistance and hyperinsulinemia may inhibit the secretion of nesfatin-1. There are no studies assessing nesfatin-1 levels dynamically in a normal pregnancy or any studies measuring first-trimester levels in women who will develop GDM. Nesfatin-1 has been shown to be an important component of the glucose dysregulation pathway in both GDM and T2DM, but the exact mechanisms and the exact cutoff values required for accurate interpretation require substantial future studies.

#### 4.3.3. Pregnancy-Associated Plasma Protein A (PAPP-A)

PAPP-A is a zinc-binding matrix metalloproteinase secreted by the trophoblast and can be measured as early as 28 days of pregnancy [281]. PAPP-a has been used as a screening test in the first-trimester of pregnancy for aneuploidy and for identifying certain adverse pregnancy outcomes [282,283]. Through its properties, PAPP-A increases insulin-like growth factor 1 (IGF-1) bioavailability through its cleavage from the IGF binding protein-4, suggesting a possible link between PAPP-A and insulin sensitivity. Pellitero et al. [284] found lower PAPP-A levels in diabetic patients compared to controls with a negative association between PAPP-A levels and HbA1c. In addition, it has been documented that TNF-α (an inflammatory cytokine with a role in insulin resistance) strongly stimulates PAPP-A secretion [285]. Therefore, it has been hypothesized that low PAPP-A levels are linked to insulin resistance in pregnancy through low levels of IGF-1 that led to hyperinsulinemia.

Lovati et al. [45], in a case–control study, explored the association of first-trimester levels of PAPP-A and GDM development in 673 Caucasian pregnant women (GDM n = 307, 100 g 3 h OGTT/ 75 g 2 g OGTT) and found significantly lower levels in women, who were diagnosed with GDM and even lower levels in women that required insulin therapy. PAPP-A (in addition to clinical risk factors) could predict GDM development with a sensitivity of 81.4%, specificity of 50.5% and an AUC of 0.70 (95% CI 0.66–0.73). Similar results are found by Ramezani et al. [46], who prospectively measured first-trimester PAPP-A in 286 Middle Eastern women (GDM n = 45, OGTT, IADPSG criteria) and found significantly lower PAPP-A levels in women who developed GDM. PAPP-A could identify the future development of GDM with a sensitivity of 73.3%, specificity of 57.3% and AUC of 0.61. This study, however, did not record certain variables, and adequate adjustments in the analysis have not been made. A similar design study [47] in a comparative size Asian cohort (GDM n = 45, OGTT, IADPSG criteria) found that PAPP-A could predict GDM with a sensitivity of 72.5%, specificity of 82.3% and AUC of 0.86.

In a large retrospective study, Snyder et al. [48] studied clinical and biomarker models for early GDM diagnosis in 66,687 (GDM n = 4874) in ethnically and racially diverse pregnant women. Samples were collected in both the first and second-trimesters of pregnancy. The team found significantly lower levels of PAPP-A in women who subsequently developed GDM compared to controls with no difference between groups in second-trimester samples. The addition of PAPP-A to the clinical risk prediction model only slightly improved the prediction accuracy of the model. These findings are supported by other studies [49,50], who did not find a significant change to AUC by the addition of PAPP-A to the clinical prediction model. While this was a large study, it only included nulliparous women, there was no information on the GDM diagnostic method or criteria used, and there was no BMI category sub-analysis. These findings suggest that PAPP-A could be useful as the first-trimester of pregnancy predictor for GDM development with limited utility as in GDM diagnosis in the second-trimester of pregnancy.

A systematic review and meta-analysis by Donovan et al. [286] included 13 studies and 83,921 subjects (GDM n = 3786). While the study identified a high degree of heterogeneity mostly due to the analysis method, the GDM diagnostic criteria and the ethnicity of subjects involved, the overall analysis and sub-analysis identified significantly lower first-trimester PAPP-A levels in women who developed GDM compared with controls with even lower levels in GDM women diagnosed prior to 24 weeks of gestation. This correlation was not as strong in women of Asian origin. The correlation between PAPP-A levels and the degree of glucose intolerance in GDM was also highlighted by Wells et al. [287], who found lower levels of PAPP-A in women with early GDM diagnosis compared to late diagnosis and the lowest levels of PAPP-A in women diagnosed with T2DM.

Research to date has not clarified if low levels of PAPP-A promote or are rather the result of impaired glucose metabolism and insulin resistance. The reduced observed levels of PAPP-A in GDM pregnancies may reflect a defect in placentation or placental insufficiency encountered in GDM pathology. The studies, however, have consistent results with overall lower PAPP-A first-trimester levels in GDM pregnancies. Moreover, these findings are also supported by studies exploring exosomes profiles as a biomarker for GDM diagnosis [288]. Despite the variable reported predictive value across the literature, which is mostly due to patients’ characteristics, sample size and GDM diagnostic criteria variability, PAPP-A is routinely assessed in first-trimester abnormalities screening, and it may identify women at high risk for early development of GDM. Further prospective studies are required to elucidate the clinical utility of this biomarker on its own or incorporated in risk identification models.

#### 4.3.4. Retinol-Binding Protein 4 (RBP4)

RBP4 is secreted mainly by the liver and adipose tissue. Its main role is to transport retinol (vitamin A) from the liver to the peripheral tissues [289]. RBP4 also has a role in inflammation and adipose tissue dysfunction [290], in increasing hepatic glucose output, in reducing insulin signaling in the muscle and in increasing insulin resistance [291].

Jin et al. [292] measured RBP4 levels in the first and second-trimester of pregnancy in 270 women (GDM n = 135, IADPSG criteria) and found that GDM women had higher first-trimester levels of RBP4 compared to NGT women, that higher levels of RBP4 were associated with a higher risk of developing GDM and that RBP4 levels in both-trimesters were positively independently associated with markers of insulin resistance. Yuan et al. [51] measured a panel of biomarkers with the potential to diagnose GDM in 359 pregnant women (GDM n = 86, IADPSG criteria) at 16–18 weeks of gestation. RBP4 was significantly higher in women that developed GDM compared to NGT. The study also found that RPB4 (cutoff value >30.45 µg/mL) could predict the development of GDM with a sensitivity of 63.6%, specificity of 75% and AUC 0.72 (95% CI 0.64–0.79) and that RPB4/adiponectin ratio (cutoff >0.37) could predict GDM with a sensitivity of 81.8%, specificity of 75.6% and AUC of 0.80 (95% CI 0.73–0.87). A retrospective study by Du et al. [29] found that second-trimester RBP4 levels were significantly higher in the GDM group (n = 194, OGTT, IADPSG criteria) compared to NGT women with a strong association between RBP4 levels, insulin levels and HOMA-IR. The authors found that RBP4 (cutoff levels 34.84 µg/mL) can predict GDM with a sensitivity of 79.4%, specificity of 79.1% and AUC of 0.87 (85% CI 0.83–0.92).

Discordant results are found in a study by Khovidhunkit et al. [293], who measured RBP4 in 532 women (GDM n = 171, GCT/OGTT, Carpenter and Coustan criteria) between 24 and 28 weeks of gestation and found no difference in RBP4 levels between GDM and NGT women and no correlation with insulin levels and HOMA-IR. They did find a positive independent association with fasting triglycerides and weight gain in pregnancy. All the women in this study were of Thai ethnicity with a low/normal BMI, which may account for the discordant result with previous studies along with different diagnosis methods (OGTT/GCT) and different sampling gestational weeks.

Two meta-analyses [294,295] studied the link between RBP4 and GDM. Huang et al. [294] included 14 studies (GDM n = 884) and found that RBP4 levels were significantly higher in women with GDM compared to NGT women, independent of age or BMI. On subgroup analysis, however, this significant difference was only maintained for Asian populations with no difference in levels in non-Asian populations. This study had a low probability of bias but a high degree of heterogeneity; the link between RBP4 levels and GDM varied with GDM diagnostic criteria (WHO criteria—higher levels of RBP4 in GDM patients compared to controls; ADA criteria—no difference in levels between groups) and varied with different assays used for RBP4 determination. These findings are supported by a meta-analysis by Jia et al. [296], who found higher levels of RBP4 in patients of Asian ethnicity compared to controls, but not in patients of European ethnicity compared to controls. A meta-analysis by Hu et al. [295] pooled results from 14 studies (case–control) (GDM n = 647) and found that RBP4 levels taken between 24 and 28 weeks of gestation were associated with the risk of developing GDM. Similar to the previous meta-analysis, there was no difference between groups in studies that used the ADA criteria for GDM diagnosis, suggesting that higher glucose levels on the OGTT are associated with higher RBP4 levels.

There are contradictory findings in the literature, with some studies finding a positive link between RBP4 and GDM [293,297,298,299,300] (Asian population), while other studies could not determine an association [301,302,303,304] (non-Asian population), and there is scarce evidence on the first-trimester of pregnancy RBP4 and the risk of GDM. If ethnicity plays such an important role in RBP4 levels, this requires further evaluation in much larger studies with multi-ethnic participation.

It is unclear if the free or bound RBP4 (or total) serves as a better predictor for GDM. Some studies [302,305] suggest that, in fact, the RBP4/transthyretin (RBP s binding protein) ratio may be a better marker for insulin resistance and GDM compared to RBP4 alone.

A possible cause for discordant results is the different assays used in the measurement of RBP4. Graham et al. [306] measured RBP4 levels in subjects with insulin resistance and glucose intolerance and insulin-sensitive subjects with NGT using three commercial assays and a quantitative Western blotting assay and found substantial inconsistency among the results with enzyme immunoassays underestimating RBP4 levels concluding that Western blotting is the most reliable method for measuring RBP4.

## 5. Discussion

This scoping review highlighted the large number of biomarkers described in the literature investigated for their potential to identify GDM and described 15 protein biomarkers selected based on the higher number of citations in very recent publications in our literature search.

### 5.1. The Current Screening Methods for GDM Pose Several Issues

#### 5.1.1. Universal vs. Selective Screening

Numerous studies [307,308,309,310] have shown that universal screening offers a significant advantage over selective screening by identifying all GDM cases, enabling timely lifestyle interventions and treatment, leading to a reduction in GDM associated adverse events. Proponents of selective screening invoke reduced complications associated with milder cases of GDM that would be diagnosed through universal screening as the argument in addition to the increased healthcare costs of screening. It is well known that GDM poses a long-term threat to the health of the mother and child through chronic metabolic diseases in a young population that will considerably increase the lifetime overall healthcare costs. Identifying GDM and intervening to prevent long-term issues has been shown to be cost-effective. A systematic review by Mo et al. [311] included 10 economic evaluations on different GDM screening strategies and found that universal screening is more likely to be cost-effective compared to selective screening. This finding is supported by other studies [312,313].

Beyond the costs, the focus should be on the accurate identification of causes and the prevention of adverse outcomes. In a Malaysian population, Idris et al. [307] found that when universal screening was employed, the OGTT yielded a sensitivity of 83.5% and specificity of 82.6%. When the selective screening was employed, the sensitivity and specificity of the OGTT were lower, 76.1% and 60.9%, respectively, leading to 23.8% of women with GDM being missed. In a European cohort, Miaihle et al. [314] showed that selective screening would have missed one-sixth of GDM cases. Similar results were found by Cosson et al. [315] in a retrospective study, which included 18,775 pregnancies. The authors found that applying selective screening criteria would lead to 34.7% of GDM cases being missed.

It was suggested that low-risk women with GDM would have a good prognosis, and not being diagnosed with GDM would not lead to adverse pregnancy outcomes. The literature has conflicting data, with some studies finding no benefit on adverse pregnancy outcomes when universal screening was applied [314,316], while others have shown significant benefits [317,318,319].

As selective screening would miss a significant number of women with GDM, and as universal screening has been shown to be cost-effective compared to selective screening, most international bodies now recommend universal screening. One of the barriers to the implementation of universal screening is the logistics of performing OGTT in the entire pregnant population. Accurate biomarkers as an alternative to the OGTT would allow universal screening to become a reality.

#### 5.1.2. Time of Screening

Standard GDM screening occurs between 24 and 28 weeks of gestation. This arguably leaves a very narrow window for intervention. Some studies suggest that women that develop GDM early in pregnancy (<12 weeks of gestation) have outcomes comparable to women with prepregnancy diabetes despite treatment [320,321], while others found that early diagnosis and treatment may lead to a reduction in LGA [322]. Research has focused on the differences in pregnancy outcomes between women with GDM diagnosed in the first and late second-trimester, but we also need to consider the possible long-term impact of the fetal intrauterine exposure to hyperglycemia between onset and diagnosis and, while this may not be obvious at birth, the hyperglycemia-triggered fetal metabolic programming can lead to metabolic syndrome, insulin resistance and obesity in young adults [323]. One of the concerns with early screening is that while some women with more severe forms of GDM will be diagnosed early in pregnancy, others will only develop glucose abnormalities later in pregnancy and will require another test at 24–28. A single non-fasting biomarker would allow repetitive testing through pregnancy to facilitate rapid identification of hyperglycemia.

#### 5.1.3. Diagnostic Criteria

As illustrated, it is difficult to make a meaningful comparison between studies when different criteria are employed to diagnose GDM. Studies that use a two-step approach with higher glucose cutoff thresholds will select a population with more severe forms of GDM, and the results cannot be extrapolated to the general population. Using a common new diagnostic biomarker will lead to harmonization of GDM diagnosis and all data synthesis to be performed

#### 5.1.4. OGTT

Women are currently diagnosed with GDM using an OGTT. This test is unreliable with poor reproducibility and high vulnerability to external and internal factors [14]. More so, the OGTT does not identify the continuous correlation between hyperglycemia in the mother and pregnancy complications and possibly omits milder forms of glucose abnormalities that may identify pregnancy risks. Studies to date evaluating novel biomarkers as diagnostic tests/tools in GDM use the OGTT as the gold standard for comparison. However, how valid are the results if the comparator test is flawed? It may be more accurate to assess the predictive power of the biomarker to identify adverse pregnancy outcomes. Hyperglycemia is not the only contributor to adverse pregnancy outcomes, and other metabolic factors, such as adiposity, dyslipidemia, inflammation, should be considered when considering a test (or panel of tests) with the highest potential to identify pregnant women at risk. Changing the focus from the glucose value to the outcomes we need to prevent—and the factors contributing to them—will bring the research community closer to identifying the next screening test. This would start with a consensus on what are the outcomes we are aiming to prevent (macrosomia, LGA, hypertensive disorders of pregnancy, polyhydramnios, etc.), a reflection on the pathophysiology of the outcome and a reassessment of novel biomarkers not in their capacity to identify an out-of-range glucose value but in their capacity to capture the cumulus of mechanisms that lead to adverse pregnancy outcomes.

The COVID-19 pandemic has highlighted the difficulties in performing the OGTT in such a challenging environment, leading to OGTT screening being terminated and a high number of women being undiagnosed. This further emphasizes the urgent need for a single non-fasting sample biomarker test that could be performed in a family practice setting (GP) rather than a hospital setting.

Identifying a biomarker for the accurate diagnosis of GDM would have numerous practical benefits. A single blood test would reduce the appointment length, would enable a greater number of women to be screened (aiming for universal screening) and would enable the test to be performed in a non-hospital setting. A test that does not require fasting would not only considerably reduce the discomfort a pregnant woman experiences but would also enable appointments for sample collection throughout the day, thus increasing the number of women being screened. A test that does not require glucose loading, reducing adverse experiences, such as nausea, vomiting, pre-syncopal episodes, would considerably increase compliance with testing. Studies assessing the robustness of novel biomarkers to pre-analytical and analytical variables, time-to-result analysis and cost-effectiveness analysis will be required in the future.

Biomarker research has grown exponentially in recent years out of a need for more accurate, more direct measurement of disease and has proven to be a powerful tool in the understanding of physiology and pathophysiology. However, while biomarkers have many advantages, much like other tests, several things need to be considered and assessed when conducting biomarker research: (1) interindividual variability; (2) intraindividual variability; (3) sample collection/transportation/storage; (4) biomarker validity; (5) predictive power; (6) confounding variables; (7) normal ranges and (8) cost [324].

The main limitation of this study lies in the nature of its design. Scoping reviews do not formally evaluate the quality of evidence, and the evidence is collected from studies of different designs and methodology. Therefore, the data collected cannot be presented in a systematic way; but instead, it gives an overall view of the existent literature. The scope of this review was to offer the reader an insight into the vast number of molecules studied in relation to GDM diagnosis and the potential diagnostic value of the selected novel protein biomarkers. This approach, however, led to a certain degree of selection bias. Another limitation of the study is the time lapse between the systematic review search and the publication of the manuscript leading to very recent research not being included in our study.

### 5.2. Perspectives

Given the limitation mentioned above, several protein biomarkers did not fulfill our inclusion criteria and were not included in the discussion. Therefore, we want to give a short overview of promising protein biomarkers evaluated for GDM prognosis/diagnosis in articles published between January 2020 and March 2021 that should be considered in the future.

Secreted frizzled-related protein 4 (sFRP4) has been shown to play a role in glucose metabolism, reflecting islet inflammation and impaired insulin secretion [325]. Schuitemaker et al. [326] found significantly higher first-trimester levels of sFRP4 in women, who subsequently developed GDM (n = 50, diagnostic criteria fasting glucose ≥ 7.0 mmol/L, 2 h glucose ≥ 7.8 mmol/L) compared to controls and a predictive capacity expressed as AUC of 0.60 (95% CI 0.50–0.70). The correlation between sFRP4 and GDM is supported by other studies [327,328]

Amini et al. [329] studied alfa-fetoprotein (AFP) as a predictor for GDM in the early second-trimester (14–17 weeks of gestation) in 523 pregnant women. The authors found that AFP alone could predict GDM diagnosis with a sensitivity of 70%, specificity of 93% and AUC of 0.58 (95% CI 0.51–0.62) and AFP combined with unconjugated estriol and β-hCG levels can predict GDM with a sensitivity of 95%, specificity of 86% and AUC of 0.91 (95% CI 0.87–0.98).

Wnt1-inducible signaling pathway protein-1 (WISP1) was studied by Liu et al. [330] in 313 pregnant women (GDM n = 61, 2 h 75 g OGTT, IADPSG criteria). The samples were taken at the time of the OGTT. The authors found that WISP1 levels were significantly higher in GDM patients with prepregnancy overweight or obesity compared to normoglycemic and normal-weight subjects, suggesting a possible role of this protein in the mechanisms involved in obesity-induced insulin resistance in GDM. This hypothesis was also suggested by Sahin Esroy et al. [331].

Irisin levels were measured by AL-Ghazali et al. [332] in 90 pregnant women (GDM n = 60, 2 h 75 g OGTT, IADPSG criteria) at the time of the OGTT and found significantly lower levels of irisin in women with GDM compared to controls. Diagnostic capacity (i.e., AUC) was not calculated. These findings are supported by other studies [333,334]

Asprosin levels [335] were found to be significantly higher in women with GDM compared with controls at the time of OGTT, but also as early as 18–20 weeks of gestation, suggesting a potential role as an early biomarker. Similarly, spexin and subfatin levels [336] and fibrinogen-like protein 1 (FGL-1) [337] were found to be higher in women with GDM compared with controls (samples taken at the time of the OGTT).

Finally, coiled-coil domain-containing 80 (CCDC80) levels [338] and complement C1q tumor necrosis factor-related protein 1 (CTRP1) levels [339] taken at the time of the OGTT (2 h 75 g OGTT, IADPSG criteria) were significantly lower in women with GDM compared with controls. Additionally, CCDC80 could identify GDM cases with an AUC of 0.61 (95% CI 0.53–0.68), which increased to 0.74 when additional variables were included in the model (maternal age, gestational age, BMI, blood pressure).

## 6. Conclusions

This review has identified and described 15 promising biomarkers that could potentially replace the OGTT and be used to both predict and diagnose GDM. Steps required to move the biomarker agenda forward should include large multicenter, multi-ethnic prospective studies using uniform screening and diagnostic criteria for GDM, with longitudinal sampling in all three trimesters and with well-recorded patient characteristics. One such study would answer many questions and help identify the best candidate marker. Recently, the Lames Lindt Alliance (Priority Setting Partnerships) has identified the top 10 research priorities for GDM, one of which is identifying the best test to diagnose GDM [340]. The scientific community agrees that the OGTT is a dated, cumbersome, imperfect test and needs to be replaced, and this review highlights some very promising contenders.

## Figures and Tables

**Figure 1 jcm-10-01533-f001:**
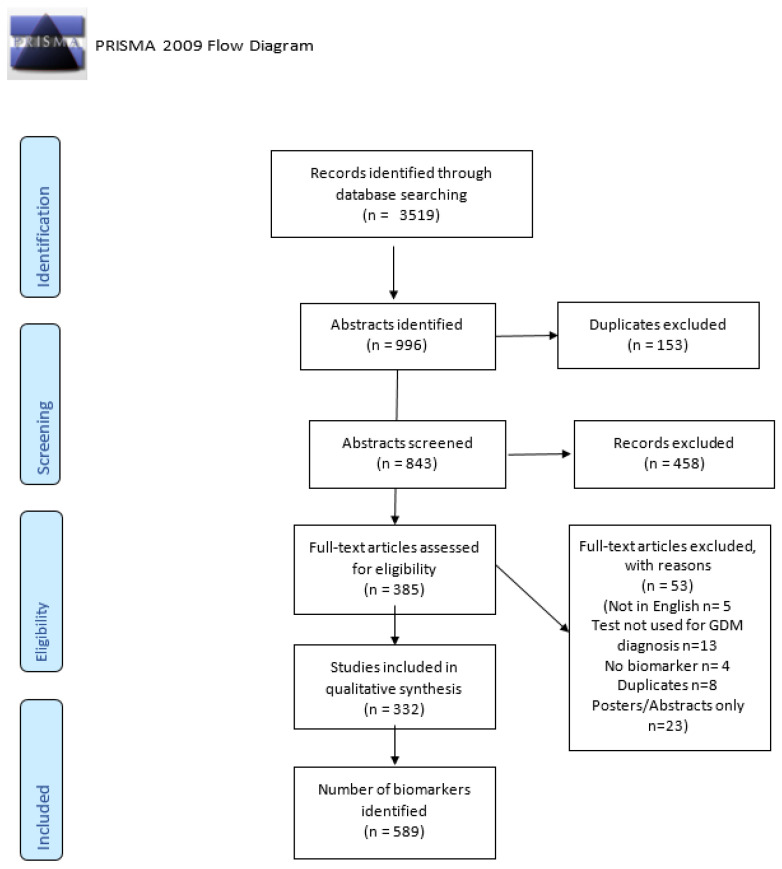
Prisma diagram.

**Table 1 jcm-10-01533-t001:** Protein biomarkers (n = 15) identified post-application of post hoc criteria.

Biomarker	Function	Molecular Characteristics
Cytokines		
* Adiponectin	Regulation of glucose and lipid metabolism. Role in cell apoptosis, inflammation and angiogenesis	Molecular mass 30 kDa; consists of 244 aa in multimeric circulating forms: a 90-kDa low molecular weight trimer, a middle molecular weight hexamer of 180 kDa and a high molecular weight multimer of ~360 kDa
* Chemerin	Adipogenesis regulation and adipocyte metabolism; role in glucose and lipid metabolism; pro/anti-inflammatory modulator	Molecular mass of 18 kDa; chemerin is translated as a 163 aa preproprotein that is secreted as a 143 aa (18 kDa) proprotein following proteolytic cleavage of a signal peptide
* Fetuin	Transport of fatty acids in the circulation with a role in insulin resistance; inhibits vascular calcification; role in inflammatory responses	Molecular mass of 64 kDa; comprises a two-chain form whose N-terminal heavy chain (321 amino acid residues) is disulfide bonded to the C-terminal light chain (27 aa)
* Leptin	Regulation of food intake and energy balance	Molecular mass of 16 kDa; consists of 146 aa structured in four antiparallel α-helices
* Omentin	Role in glucose and lipid metabolism and adipocyte mediated inflammation	Molecular mass of 34 kDa; consists of 313 aa; contains a secretory signal sequence and a fibrinogen-related domain and appears as a glycolyzed trimer of 120 kDa molecular weight in its negative form
IL-6	Role in immunity as a mediator of the acute phase response. Acts as both a proinflammatory cytokine and an anti-inflammatory myokine. Additional role in adipocyte-mediated inflammation and glucose metabolism	Molecular mass of 21 kDa; single, non-glycosylated polypeptide chain with a four–α-helix structure containing 185 aa
TNF	Role in the regulation of immune cells, growth regulation, inflammation, viral replication, tumorigenesis, and autoimmune diseases	Molecular mass of 17.3 kDa; homotrimer composed of 233 aa
Glycoproteins		
Afamin	Vitamin E transport. Possible role in glucose and lipid metabolism	Molecular mass of 87 kDa with 55% aa sequence similarity to albumin; composed of a 21-aa leader peptide, followed by 578aa of the mature protein and consists of 2 structural domains
hCG	Maintains the production of progesterone from the corpus luteum during pregnancy; role in glucose and insulin metabolism; role in adipocyte-mediated inflammation	Molecular mass of 36.7 kDa, (~14.5 αhCG and 22.2 kDa βhCG), composed of 237 aa; it is heterodimeric, with an α subunit identical to that of luteinizing, follicle-stimulating and thyroid-stimulating hormone and an β subunit that is unique to hCG
CD59	Inhibits the complement membrane attack complex action	Molecular mass of 14.2 kDa; consists of 128 aa
SHBG	Binding protein for testosterone and estradiol; regulates sex steroid effects in target cells by direct action; role in lipid and glucose metabolism	Molecular mass of 43.7 kDa; homodimer, each monomer consists of 402 aa
Other Proteins		
CRP	Activation of the complement system, promoting phagocytosis by macrophages Role in the innate immune system	Molecular mass of 120 kDa, belonging to the family of pentraxins; consists of five identical subunits that contain each 206 aa
Nefatin-1	Regulation of food intake and glucose homeostasis	Molecular mass of 9.7 kDa containing 82 aa residues
PAPP-A	Cleavage of insulin-like growth factor-binding proteins promoting somatic growth	Molecular mass of 400 kDa composed of two 200-kDa disulfide-bound subunits, each subunit consists of 1547 aa: belonging to the pappalysin protein family
RBP4	Transporter protein for retinol; role in insulin resistance and tumor growth	Molecular mass of 21 kDa consisting of 184 aa; the entire molecule consists of an N-terminal loop, a β-barrel structure, an alpha helix and a C-terminal loop

* adipokines; IL-6—interleukin 6; TNF—tumor necrosis factor; hCG—human chorionic gonadotropin; SHBG; sex hormone-binding globulin; CRP—C-reactive protein; PAPP-A—placental associated plasma protein A; RBP4—retinol-binding protein 4; aa—amino acids.

**Table 2 jcm-10-01533-t002:** Summary of test performance at the time of gestational diabetes (GDM) diagnosis *.

Biomarker	First Author (Ref.)	Analytical Method	Diagnostic Sensitivity %	Diagnostic Specificity %	AUC	Cutoff Value
Cytokines						
Adiponectin	Bozkurt et al. [22]	RIA	NS	ns	0.62	ns
	Weerakiet et al. [23]	ELISA	91.7	30.8	0.63	10 µg/mL
Chemerin	Wang et al. [24]	ELISA	73.3	76	0.82	6.78 µg/L
Leptin	Bozkurt et al. [22]	RIA	ns	ns	0.61	ns
	Boyadzhieva et al. [25]	ELISA	81.2	64.2	0.82	28.7 ng/mL
Glycoproteins						
CD59	Ghosh et al. [26]	ELISA	85	92	0.92	ns
	Ma et al. [27]	ELISA	54	93	0.86	ns
SHBG	Tawfeek et al. [28]	ELISA	96	95	0.91	50 nmol/L
Other Proteins						
RBP4	Du et al. [29]	ELISA	79.4	79.1	0.87	34.84 µg/mL)

* no information on test performance at the time of GDM diagnosis was found for the following biomarkers: fetuin, omentin, IL-6, TNF, afamin, hCG, CRP, nesfatin-1, PAPP-A. AUC—area under the curve; IL-6—interleukin 6; TNF—tumor necrosis factor; hCG—human chorionic gonadotropin; SHBG—sex hormone-binding protein; CRP—C-reactive protein; PAPP-A—pregnancy-associated plasma protein A; RBP4—retinol-binding protein 4; RIA—radioimmunoassay; ELISA—enzyme-linked immunosorbent assay; ns—not stated.

**Table 3 jcm-10-01533-t003:** Summary of test performance as a predictive indicator of GDM *.

Biomarker	First Author, Year (Ref.)	Analytical Method	Diagnostic Sensitivity %	Diagnostic Specificity %	AUC	Cutoff Value
Cytokines						
Adiponectin	Georgiou et al. [30]	ELISA	85 ^1^	85.7 ^1^	0.86	3.5 µg/mL
	Ferreira et al. [31]	ELISA	ns	ns	0.85 ^2^	ns
	Madhu et al. [32]	ELISA	100	95.6	ns	9.1 µg/mL
	Iliodromiti et al. [33] **	ns	64.7	77.8	0.78	ns
Fetuin	Kansu-Celik et al. [34]	ELISA	58.6	76.2	0.33	166 ng/mL
	Jin et al. [35]	ELISA	64.4	58.5	0.61	305.9 pg/mL
Leptin	Bawah et al. [36]	ELISA	95.7	68.6	0.81	18.9 ng/mL
TNF	Syngelaki et al. [37]	ELISA	ns	ns	0.82	ns
Glycoproteins						
Afamin	Tramontana et al. [38]	ELISA	ns	ns	0.66 ^3^	ns
	Koninger et al. [39] **	ELISA	79.3	79.4	0.78	88.6 mg/L
	Ravnsborg et al. [40] **	nanoLC-MS	ns	ns	0.67	ns
SHBG	Caglar et al. [41]	RIA	46.7	84.1	0.87	97.47 nmol/L
	Maged et al. [42]	ELISA	85.2	37	0.69	211.5 nmol/L
	Veltman-Verhulst et al. [43] **	ECL	81	82.8	0.86	58.5 nmol/L
	Badon et al. [44] **	ELISA	ns	ns	0.71 ^2^	44.2 nmol/L
Other Proteins						
CRP	Kansu-Celik et al. [34]	Nephelometry	86.2	50.8	0.70	ns
PAPP-A	Lovati et al. [45]	DELFIA	81.4 ^2^	50.5 ^2^	0.70 ^2^	ns
	Ramezani et al. [46]	ELISA	73.3	57.3	0.61	1896 mU/L
	Ramezani et al. [46]	ELISA	34.4	83.2	0.62	0.3 mU/L
	Ren et al. [47]	TRFIA	72.5	82.3	0.86	16.34 ng/L
	Snyder et al. [48]	DELFIA	75.7 ^2^	55.5 ^2^	0.71 ^2^	ns
	Xiao et al. [49]	DELFIA	ns	ns	0.53; 0.68 ^2^	ns
	Syngelaki et al. [50]	DELFIA	ns	ns	0.84 ^2^	ns
RBP4	Yuan et al. [51]	EIA	63.6	75	0.72	30.45 µg/mL

* no information on test performance as a predictive indicator of GDM was found for the following biomarkers: chemerin, omentin, IL-6, CD59, hCG, nesfatin-1. AUC—area under the curve; IL—6—interleukin 6; TNF—tumor necrosis factor; hCG—human chorionic gonadotropin; SHBG—sex hormone-binding protein; CRP—C-reactive protein; PAPP-A—pregnancy-associated plasma protein A; RBP4—retinol-binding protein 4; RIA—radioimmunoassay; ELISA—enzyme-linked immunosorbent assay; nanoLC-MS—nano-flow liquid chromatography-tandem mass spectrometry; ECL—electrochemiluminescence; DELFIA—dissociation-enhanced lanthanide fluorescent immunoassay, TRFIA—time-resolved fluorescence immunoassay analyzer; EIA—enzyme immunoassay; ns—not stated; ^1^ combined model with insulin levels; ^2^ combined model with risk factors; ^3^ combined model with BMI; ** prior to pregnancy.

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
