# Peer review of "Emerging Protein Biomarkers for the Diagnosis or Prediction of Gestational Diabetes—A Scoping Review"

_jcm, 2021, doi:10.3390/jcm10071533_

Round 1
Reviewer 1 Report
Thank you for a great manuscript!
February 2021, Review for the Journal of Clinical Medicine of the manuscript:
Emerging protein biomarkers for the diagnosis or prediction
of gestational diabetes – a scoping review
Outline
Scoping review including 332 full-text reviewed articles.
Identification of 589 biomarkers in relation to GDM diagnosis, hereof 15 selected from 3 post-hoc criteria: 1) protein biomarkers only, 2) minimum 5 citations, and 3) published between 2017-2020.
Aim: To synthesize the literature on emerging biomarkers for GDM diagnosis (and screening).
General comments
- A very important and highly relevant topic described in this thorough and comprehensive work – congratulations on a great piece of work.
- The methodological procedure including literature search is carefully conducted and adhere to the expected high standards apart from the use of post-hoc selection.
- The work would be advanced to an even higher level by changing the structure and expand the discussion (please see below).
Major concerns
- I find it somewhat troublesome that of 589 identified biomarkers only 15 were selected for review. This rather coarse sorting resulting from the 3 applied post-hoc criteria, may induce important bias including framing -, selection -, confirmation -, reporting - and observe-expectancy bias (which is not discussed). For example, a biomarker may not have been cited at least 5 times, due to the fact that it is a very newly discovered biomarker. In this case, it could be of great value to include a small paragraph on the ‘newest identified promising biomarkers within the last year’ (‘Perspectives’, please see below).
- Furthermore, a common problem in research on biomarkers is that they may show promising results in the first few studies but then in following studies with smaller cohorts, the effect diminishes. These challenges could be highlighted in a small paragraph on ‘pros et cons’ in biomarker research.
- Structure: It works really well dividing the biomarkers in sub groups of cytokines, glycoproteins and other proteins, however, I miss a more systematic approach. I would suggest a tight systematic structure when describing the biomarkers, for example making an overview of the studies measuring the biomarkers in first vs second trimester, as well as a concluding paragraph for each biomarker, summarizing the findings, pros et cons etc.
- Regarding the discussion/conclusion, this could be given a much greater role in the manuscript, since this is - in my opinion - the most interesting part. It could also be considered to expand with a ‘Perspectives’ section. In particular, two topics in the discussion may be elaborated:
- Universal vs selective screening
It would be lovely if this paragraph could give an even more detailed yet critical overview of ‘pros et cons’ for universal vs selective screening.
- The OGTT as golden standard for GDM diagnosis
This topic in particular could be given more attention since it is - to me - the very core of the discussion: Which outcomes should be used as the measure of significant problems; problems big enough to justify the diagnosis of GDM? Using the OGTT, we only look at hyperglycemia but what about other important contributing parameters that result in adverse outcomes? Perhaps it would be of greater gain to look for biomarkers detecting, for example, LGA. Or to use the outcomes from the HAPO study (e.g. birth weight >90th percentile, cord-blood serum C-peptide >90th percentile, and infant body fat >90th percentile) that were the basis for the OGTT diagnostic cut-offs recommended by the IADPSG.
Minor concerns
- Overall, the manuscript contains some typos, as well as grammatical and linguistic errors. Therefore, a thorough proofreading is recommended also regarding consistency in the use of scientific/clinical terms or definitions throughout the manuscript (e.g. the use of either ‘WHO 2013 criteria’ or ‘IADPSG criteria’, ‘weeks of gestation’ or ‘weeks of pregnancy’, and ‘first trimester’ or ‘trimester 1’).
- Some sentences/paragraphs could perhaps be rephrased or shortened down, since they are slightly difficult to read:
- 5 line 9-11*
- 7 line 40
- 8 line 24-25 and 32-33
- 9 line 30-32 and 33-34
- 10 line 6-7
- 11 line 2-3
- 15 line 21-23
- 16 line 19-22
- 18 line 22-26 and 44-45
- 21 line 41 - p. 22 line 1
*lines are calculated including headings and excluding section gaps
- 11 line 28-30: It is not clear to me how you can conclude that age and BMI might be important parameters influencing omentin levels from the finding: ‘…omentin was significantly lower in women with GDM…’. Perhaps you could make this clearer?
- There is a repetition of a paragraph in p. 4 line 29, and p. 5 line 5.
- Consider changing the phrase ‘diabetic subject’ (p. 15) to ‘women with diabetes’.
Comments on supplemental material
- The tables 2 and 3 give a great overview, which could also be achieved in the text if changing the structure (please see above). Could a similar overview - obviously using other parameters than sensitivity/specificity/AUC - be made for the 4 biomarkers not mentioned in these tables (i.e. omentin, IL-6, hCG and Nesfatin-1)?
- It is excellent with the alphabetic list of all biomarkers identified from the literature search (suppl. table 1).
- Would it be possible to create a graphic illustration of a hierarchy of the reviewed biomarkers, ranking them as ‘the best/most promising’ and so forth? This would help provide a great overview.
Reviewer 2 Report
In the present work, Bogdanet et al. clearly report evidence of several circulating biomarkers which are mainly correlated with the prediction of gestational diabetes onset, at the expense of OGTT, that, as previously described by same authors, is highly variable and mainly dependent on several clinical an pre-analytical conditions. The work is well written although it need some language improvements. The meta-analysis is well conducted and the method through which it was carried out is well described.
However, some observation should be done:
- The structure of the review is a little bit redundant and adds almost nothing new compared to another recently published review (Predictive and diagnostic biomarkers for gestational diabetes and its associated metabolic and cardiovascular diseases; A. Lorenzo-Almorós et al. 2019).
- I cannot understand why authors decided to include papers published until July 2020 and not at least until December 2020; I think literature should be updated.
- Due to the redundancy of the review compared to the already cited one, I highly recommend to authors to completely revise the work and to describe these biomarkers from another point of view, for example by discussing how much these reproducible biomarkers are objectively applicable to daily clinical practice.
Round 2
Reviewer 2 Report
The work has been significantly improved. I would just like to suggest to improve Table 1 by adding also in the table the function and some molecular charachteristic of the described biomarker.
